# Dysphagia Prevalence, Time Course, and Association with Probable Sarcopenia, Inactivity, Malnutrition, and Disease Status in Older Patients Admitted to an Emergency Department: A Secondary Analysis of Cohort Study Data

**DOI:** 10.3390/geriatrics6020046

**Published:** 2021-04-26

**Authors:** Tina Hansen, Rikke Lundsgaard Nielsen, Morten Baltzer Houlind, Juliette Tavenier, Line Jee Hartmann Rasmussen, Lillian Mørch Jørgensen, Charlotte Treldal, Anne Marie Beck, Mette Merete Pedersen, Ove Andersen, Janne Petersen, Aino Leegaard Andersen

**Affiliations:** 1Department of Occupational Therapy and Physiotherapy, Copenhagen University Hospital Amager and Hvidovre, 2650 Hvidovre, Denmark; 2Department of Clinical Research, Copenhagen University Hospital Amager and Hvidovre, 2650 Hvidovre, Denmark; rikke.lundsgaard.nielsen@regionh.dk (R.L.N.); morten.baltzer.houlind@regionh.dk (M.B.H.); juliette.tavenier@regionh.dk (J.T.); line.jee.hartmann.rasmussen@regionh.dk (L.J.H.R.); lillian.moerch.joergensen@regionh.dk (L.M.J.); ctreldal@gmail.com (C.T.); mette.merete.pedersen@regionh.dk (M.M.P.); ove.andersen@regionh.dk (O.A.); janne.petersen.01@regionh.dk (J.P.); aino.leegaard.andersen@regionh.dk (A.L.A.); 3Department of Clinical Medicine, Faculty of Health and Medical Sciences, University of Copenhagen, Blegdamsvej 3B, 2200 Copenhagen N, Denmark; 4The Capital Region Pharmacy, Marielundsvej 25, 2730 Herlev, Denmark; 5Department of Drug Design and Pharmacology, University of Copenhagen, Universitetsparken 2, 2100 Copenhagen Ø, Denmark; 6Department of Psychology and Neuroscience, Duke University, 2020 W Main St, Durham, NC 27705, USA; 7Emergency Department, Copenhagen University Hospital Amager and Hvidovre, Kettegaards alle 30, 2650 Hvidovre, Denmark; 8Department of Nursing and Nutrition, University College Copenhagen, Sigurdsgade 26, 2200 Copenhagen N, Denmark; Anne.Marie.Beck@regionh.dk; 9Dietetic and Nutritional Research Unit, Herlev-Gentofte University Hospital, Borgmester Ib Juuls Vej 50, 2730 Herlev, Denmark; 10Center of Clinical Research and Prevention and Department of Clinical Pharmacology, Copenhagen University Hospital Bispebjerg and Frederiksberg, Nordre Fasanvej 57, 2000 Frederiksberg, Denmark; 11Section of Biostatistics, Department of Public Health, University of Copenhagen, Øster Farimagsgade 5, 1014 Copenhagen K, Denmark

**Keywords:** swallowing difficulties, sarcopenia, inactivity, malnutrition, acute care, geriatric patients

## Abstract

There is evolving evidence for an association between dysphagia and sarcopenia in older adults. For optimizing the acute health care initiative across health care settings, this study investigated prevalence and time-course of dysphagia in older patients admitted to an emergency department (ED) as well as its association with parameters for probable sarcopenia, inactivity, malnutrition, disease status, and systemic inflammation. A secondary analysis of data from the FAM-CPH cohort study on acutely admitted older medical patients (*n* = 125). Data were collected upon ED admission as well as four and 56 weeks after discharge. Using the Eating Assessment Tool cut-off score ≥ 2, signs of dysphagia were present in 34% of the patients at ED admission and persisted in 25% of the patients 56 weeks after discharge. Signs of dysphagia at 56-week follow-up were significantly (*p* < 0.05) associated with probable sarcopenia (low handgrip strength (OR = 3.79), low leg muscle strength (OR = 8.14), and low physical performance (OR = 5.68)) and with baseline swallowing inactivity (OR = 5.61), malnutrition (OR = 4.35), and systemic inflammation (OR = 1.33). Signs of dysphagia in older patients admitted to an ED was prevalent, persisted 56 weeks after discharge, and was associated with probable sarcopenia and related conditions; all modifiable targets for management of dysphagia in older patients.

## 1. Introduction

Dysphagia in older adults is a frequent and serious condition that impairs swallowing efficiency and safety with increased risk of diminished nutritional intake and aspiration of foods and liquids [1,2]. Consequently, dysphagia is associated with aspiration pneumonia, malnutrition, weight loss, frequent hospital admissions with prolonged length of stay (LOS), increased mortality, decreased quality of life, as well as increased healthcare costs [1,3,4,5,6,7]. The prevalence of dysphagia in geriatric patients is reported as high as 50% [5,6,7] and can be caused by a range of diseases of the central and peripheral nervous system, structural, and mechanical changes of the aerodigestive tract as well as by advanced age [1,2,8].

There is evolving evidence for an association between sarcopenia (low muscle strength, low muscle quantity or quality, and low physical performance) [9] and dysphagia in older adults [8,10,11]. Concurrent dysphagia and sarcopenia are observed in 6% of community-dwelling elders [12], in 13–30% of geriatric inpatients [13,14,15], and in 32% of patients requiring dysphagia rehabilitation [16]. Sarcopenia of the whole body and swallowing muscles can be primary due to aging per se, or secondary in the event of physical inactivity, malnutrition, and/or systemic disease and inflammation [8,11,17]. The presence of concurrent dysphagia and sarcopenia is complex and there may be a two-way causal relationship between them generating detrimental synergies [8,11]. In a large retrospective cohort study, Maeda et al. [18] found that in older patients with normal swallow at admission, the risk of developing dysphagia during hospitalization was associated with indicators on sarcopenia, swallowing inactivity caused by diet restrictions with no oral intake or being on a texture modified diet, low performance status and immobilization, and low nutritional status. Likewise, dysphagia leads to reduced swallowing activity and malnutrition due to decreased food intake, which might induce or exacerbate sarcopenia [11].

Research on dysphagia and sarcopenia is still in its infancy and further research in various settings is needed [8,11]. During the last decades, the Danish healthcare system has been reorganized and centralized into fewer hospitals and a single-entry point through the emergency department (ED) with the aim to support cooperation across medical specialties and to contribute to faster treatment, reduced LOS and/or avoidance of unnecessary hospitalization [19]. In addition, the number of ED contacts and the proportion of contacts lasting < 24 h among the older population have increased, and a growing part of healthcare, treatment, and rehabilitation services are provided by the municipalities in the primary care setting [20]. Accordingly, it is recommended that the overall acute health care initiative is seen as a unified effort across sectors to ensure high quality and efficiency in the process for the acute older patient [20]. In this context, evidence of dysphagia prevalence, course, and association with sarcopenia in older patients admitted to the ED will contribute to optimizing patient trajectory from secondary health- to primary health- and social care settings. One Danish feasibility study reports a dysphagia prevalence of 24% in older patients admitted to the ED [21]. However, this study did not explicitly address the course of dysphagia or its association with sarcopenia. Therefore, the aim of the present study was to investigate the prevalence and time course of dysphagia and its association with parameters of sarcopenia, inactivity, malnutrition, disease status, and systemic inflammation in older patients acutely admitted to an ED within the Danish health care system.

## 2. Materials and Methods

### 2.1. Design and Participants

This study is a secondary analysis of data collected for a prospective, longitudinal observational study (the FAM-CPH cohort) investigating mechanisms for chronic inflammation and biological aging, risk of malnutrition, and unnecessary medication after admission to an ED at Copenhagen University Hospital, Hvidovre, Denmark (Clinical Trials.gov identifier: NCT03052192). Details on the FAM-CPH study have previously been described [22,23]. In short, eligible patients were aged ≥ 65 years, Caucasian, and had sufficient Danish language skills. Patients were excluded if unable to participate due to cognitive impairments, terminally ill, or in need of isolation due to infectious disease. Data were collected from November 2016 to September 2018 upon ED admission, at 4 weeks and 56 weeks after discharge according to a comprehensive data collection manual.

### 2.2. Data Included in the Secondary Analysis

For the present secondary analysis, data were provided from the Research Electronic Capture tool (REDCap) [24] and included information on baseline patient characteristics, dysphagia (primary outcome), and a set of secondary outcomes related to sarcopenia, activity status, nutritional status, and disease status upon ED-admission (baseline), and at 4-week and 56-week follow-up after discharge.

#### 2.2.1. Patient Characteristics

Patient characteristics at baseline includes demographics, admission diagnoses, LOS, and cognitive status assessed with the Orientation Memory Concentration-test (OMC) that provides a weighted total score ranging from 0–28 points, where scores ≤ 17 indicates moderate to severe cognitive impairment [25].

#### 2.2.2. Dysphagia

Dysphagia was assessed by the Eating Assessment Tool (EAT-10); a patient-reported outcome measure of self-perceived dysphagia severity. The EAT-10 covers ten items, each reflecting a possible sign of dysphagia. The items are rated on a five-point response scale from 0 (no problem) to 4 (severe problem) with a total score ranging from 0–40 [26]. A total score ≥ 2 has been shown as indicative of dysphagia with a sensitivity of 94.0% and a specificity of 70.9%, when using video fluoroscopic swallowing examination as reference test [27]. This cut-off was used for estimating the prevalence of signs of dysphagia at the three time points. In addition, the course of dysphagia from baseline to 4-week and 56-week follow-up was categorized as follows: dysphagia persistently absent (EAT-10 total score < 2 at all three time points); dysphagia persistently present (EAT-10 total score ≥ 2 at all three time points); dysphagia remission (EAT-10 total score ≥ 2 at baseline or at 4-week follow-up and EAT-10 total score < 2 at 4-week or at 56-week follow-up), and dysphagia incident (EAT-10 total score < 2 at baseline or at 4-week follow-up and EAT-10 total score ≥ 2 at 4-week or at 56-week follow-up).

#### 2.2.3. Sarcopenia

In the FAM-CPH dataset, available sarcopenia-related parameters were muscle strength and physical performance. According to the European Working Group on Sarcopenia in Older People revised guidelines (EWGSOP2), low muscle strength indicates probable sarcopenia and physical performance provides information on sarcopenia severity [9].

Muscle strength was determined by handgrip strength (HGS) and leg strength and endurance. HGS was assessed using a digital hand-held dynamometer (model Digi-II, Saehan Corp., Masan, South Korea). Bedridden patients were assessed in the bed with an elevated backrest, and mobile patients were sitting on a chair with the elbow flexed at 90 ° and the wrist in a neutral position [28]. The highest value of three consecutive attempts with the dominant hand is used for analyses. If the third trial elicited the highest value, additional trials were performed until maximum HGS was identified. Leg strength and endurance was assessed using the 30 second chair stand test (30-CST), which assesses how many times a patient can rise and sit from a standardized chair with the arms folded across the chest in 30 seconds. Only full standing positions are counted, and scores can range from 0 for those who cannot complete 1 stand to above 20 for more fit individuals [29]. Before data collection, patients were asked to do one or two practice repetitions to ensure they understood the expected performance. Physical performance was determined by the 4-m gait speed (4MGS) test, of which, the fastest time of two trials (expressed in meter per second (m/s)) at usual pace was used for the analyses [30].

According to suggested cut-off points, the following thresholds were used for the sarcopenia-related parameters: HGS < 27 kg for men and < 16 kg for women, 30-CST < 9 rises, and gait speed ≤ 0.8 m/s [9,31]. In the FAM-CPH dataset, information on muscle quality and quantity was not provided. According to the EWGSOP2 [9], a confirmed diagnosis of sarcopenia includes evidence of low muscle quality and quantity. Therefore, a definite diagnosis of sarcopenia as defined by EWGSOP2 [9] was not possible, and the three included sarcopenia parameters were considered individually as signs of probable sarcopenia.

#### 2.2.4. Activity, Nutritional, and Disease Status

Activity status was represented by swallowing activity and functional performance status. Swallowing activity was determined on basis of self-report using the Simplified Nutritional Appetite Questionnaire (SNAQ) with four items to be rated on a five-point Likert scale providing a total score from 4 to 20, where higher scores indicate better appetite and oral intake [32]. In present study, swallowing inactivity was defined by a SNAQ score < 14, which has been reported to be a significant risk of weight loss > 5% within 6 months with a sensitivity of 81.5% and a specificity of 76.4% [32]. Functional performance status was determined based on self-reports using the functional recovery score (FRS) with 11 items distributed into three main areas of basic activities of daily living (4 items), instrumental activities of daily living (6 items), and mobility (1 item). Each item is rated from 0 (total dependence) to 4 (total independence) and is summarized into a weighted total score from 0 (total dependence) to 100 (complete independence) [33,34].

Nutritional status was assessed by the Mini Nutritional Assessment Short Form (MNA-SF) and the body mass index (BMI). The MNA-SF comprises six items related to food intake, unintentional weight loss, neuropsychological problems, acute disease, and mobility, and one item measuring BMI, which is calculated as body weight (kg)/squared body height (m^2^) (reference range, 18.5–24.9). The MNA-SF score ranges from 0 to 14 points, where 12–14 points reflect normal nutritional status, 8–11 points reflect risk of malnutrition, and 0–7 points reflect malnutrition [35].

Disease status was represented by degree of comorbidity burden and chronic inflammation. Comorbidity burden was determined upon the medical anamnesis of additional diagnoses (ICD-10) extracted from patients’ medical charts and scored according to the age-adjusted Charlson Comorbidity Index (CCI) with a score range of 0 to 43 [36]. Chronic inflammation was represented by four inflammatory biomarkers given their previous association with sarcopenia in older adults [37,38]: C-reactive protein (CRP) (reference range < 10 milligram per liter (mg/L)), soluble urokinase plasminogen activator receptor (suPAR) (reference range < 3 nanogram per milliliter (ng/mL)), tumor necrosis factor (TNF)-α (reference range ≤ 8.1 picogram per mL (pg/mL)), and interleukin (IL)-6 (reference range < 16 pg/mL) [37,38]. The applied measurement methods of these biomarkers are described in detail in Tavenier et al. [23].

### 2.3. Statistics

The secondary analyses were performed as complete case analyses. Initial inspections of the dataset with 125 patients who completed the EAT-10 at baseline revealed that 48% (*n* = 60) were lost at 56-week follow-up (Appendix A). In addition, there was > 20% missing values at baseline for the 30-CST and the 4MGS (Appendix A). Given that the data were not missing completely at random (Appendix A) and that most of the variables were non-normally distributed as measured by the Kolmogorov–Smirnov test, imputation was not applied [39].

Descriptive statistics were used to summarize patient characteristics, prevalence of signs of dysphagia, and the secondary outcomes. Categorical variables were summarized as percentages, quantitative variables normally distributed as mean and standard deviation (SD), and non-normally distributed variables as median with interquartile range. Differences between patients according to swallowing status were analyzed using *t*-test for continuous data and Mann–Whitney U-test for sum-scores based on ordinal scales or continuous data non-normally distributed. For data on nominal level, continuity-corrected chi-square (χ^2^) test or Fisher’s exact test were used. The difference in dysphagia severity level across the three time points was analyzed using Friedman test. Strength of the associations between dysphagia and the parameters for probable sarcopenia as well as activity, nutritional and disease status at the three time points was also expressed as crude odds ratio (OR) with 95% confidence intervals (95% CI) [40]. Association of probable sarcopenia, activity status, nutrition, and disease status at baseline with signs of dysphagia (persistent and incident) at 56-week follow-up were investigated using logistic regression analysis. Due to the limited sample size and power, multinominal logistic regressions or multivariate models were not constructed [40]. All analyses were performed using the Statistical Package for Social Sciences (SPSS) version 25.0, and all *p*-values were two-sided with a significant level of 5%.

## 3. Results

Table 1 displays the characteristics of 125 patients who completed the EAT-10 at baseline, the patients who completed at 56-week follow-up (*n* = 65) and those loss to follow-up (*n* = 60). Reasons for loss to follow-up are presented in Appendix A. At baseline, the mean age was 78.6 ± 8.3 years, 56% were women, 88% lived in a private residence and about 60% were living alone. Patients who were loss to follow-up were older, had lower muscle strength and gait speed, poorer functional performance- and nutritional status, higher degree of comorbidity burden, higher plasma levels of the four inflammatory biomarkers, and longer LOS than patients who completed (Table 1). There was no difference in the EAT-10 scores between patients who completed and those lost to follow-up at 56 weeks.

Using an EAT-10 score ≥ 2, signs of dysphagia was present in 34% of the patients at baseline, in 24% at 4-week follow-up and in 25% at 56-week follow-up. The course of dysphagia in the subgroup of the cohort who had complete EAT-10 questionnaires at all three timepoints (*n* = 65) is illustrated in Figure 1.

At 4-week-follow-up, signs of dysphagia were persistently absent in 39 patients (61%), remitted in 10 patients (15%), were incident in 6 patients (9%), and were persistently present in 10 patients (15%). At 56-week follow-up, signs of dysphagia were persistently absent in 42 patients (64%), remitted in 7 patients (11%), were incident in 7 patients (11%) and were persistently present in 9 patients (14%). There were no significant differences in the dysphagia severity level (χ^2^ (2) = 129, *p* = 0.524) across the three time points.

As illustrated in Table 2, signs of dysphagia were related to low HGS at baseline (*p* = 0.012) and at 56-week follow-up (*p* = 0.043), low 30-CST at baseline (*p* = 0.046) and at 4-week (*p* = 0.024) and 56-week (*p* = 0.002) follow-up, and low 4MGS at 56-week follow-up (*p* = 0.021). Table 3 shows that the likelihood of presenting signs of dysphagia in the occurrence of probable sarcopenia was up to 3 times higher at baseline and 4-week follow up (ORs ranging from 1.38–3.56), and about 4 to 8 times higher at 56-week follow-up (ORs ranging from 3.79–8.14). 

Patients with signs of dysphagia had lower swallow activity and functional performance status at all three time points and showed poorer nutritional status at baseline and 56-week follow-up than patients without signs of dysphagia (Table 2). Table 3 displays that higher levels of swallow activity and functional performance and better nutritional status appeared protective for signs of dysphagia with ORs ranging from 0.81–0.98 at baseline and 4-week follow-up, and from 0.56–0.96 at 56-week follow-up. Signs of dysphagia was not related to comorbidity burden or plasma levels for the four inflammatory biomarkers, except for TNF-α levels at 56-week follow-up, which were significantly higher in patients with signs of dysphagia than in patients without (*p* = 0.039) (Table 2).

The univariate logistic regressions displayed in Table 4, indicate that low leg muscle strength and endurance (OR = 4.89, *p* = 0.030), low swallowing activity (OR = 5.61, *p* = 0.005), poor nutritional status (OR = 4.35, *p* = 0.023), and higher plasma levels of suPAR (OR = 1.33, *p* = 0.035) and IL-6 (OR = 1.08, *p* = 0.035) at baseline were associated with signs of dysphagia (persistent or incident) at 56-week follow-up. In addition, higher levels of swallow activity and functional performance and better nutritional status appeared protective for signs of dysphagia with ORs ranging from 0.68–0.95.

## 4. Discussion

The present study investigated the prevalence and time course of signs of dysphagia as well as its association with parameters for probable sarcopenia, activity, nutritional status, and disease status in older patients acutely admitted to an ED. It was found that signs of dysphagia were present in 34% of the patients at baseline, in 24% at 4-week and in 25% at 56-week follow-up. Patients with signs of dysphagia also demonstrated signs of probable sarcopenia and they had lower swallowing activity and functional performance status as well as poorer nutritional status at all three time points compared to patients without signs of dysphagia. At 56-week follow-up, patients with dysphagia had higher levels of the inflammatory biomarker TNF-α than patients without signs of dysphagia. Univariate logistic regressions revealed that signs of probable sarcopenia as well as inactivity, malnutrition, and chronic inflammation at baseline were significantly associated with signs of dysphagia at 56-week follow-up.

The dysphagia prevalence of 34% in the FAM-CPH-cohort at baseline is in line with the body of literature, in which cross-sectional prevalence estimates of dysphagia range from 30% to 50% in hospitalized geriatric patients [1,5,6,7] and in 24% of ED patients [21]. However, there are large variations in the methodology for identifying dysphagia across studies. In the FAM-CPH study, EAT-10 was the only measure of dysphagia. Formal diagnosis of dysphagia is usually carried out using a comprehensive clinical bedside examination that includes physical examination of oral and motor function, and assessment of functional oral intake level of foods and liquids, or by instrumental assessments such as video fluoroscopic swallowing examination or fiberoptic endoscopic evaluation of swallowing [41]. EAT-10 is not designed for diagnosing dysphagia per se [26]. Recent work has shown that when EAT-10 is used for identifying dysphagia in older adults, it presents with low reliability and a substantially high floor effect (i.e., no problem) and might not identify patients with milder signs of dysphagia [42]. Therefore, it cannot be excluded that the prevalence of signs of dysphagia in the FAM-CPH cohort is underestimated and the course of dysphagia is biased.

This study found that signs of dysphagia in older ED patients were associated with low muscle strength and low physical performance. However, these sarcopenia-related parameters appeared more affected at baseline than at 4-week follow-up, which might be due to the patients’ acute condition rather than a true reflection of muscle strength and physical performance at baseline [18]. In addition, it is particularly important to note that the data for this secondary analysis did not allow a definite diagnosis of sarcopenic dysphagia, which is defined as dysphagia due to sarcopenia in both generalized skeletal muscles and swallowing-related muscles [8,11]. Studies report that older patients with dysphagia and sarcopenia show low strength in tongue and lip muscles [43] and hyoid muscles [44] as well as weak pharyngeal contraction and upper esophagus sphincter dysfunction [45]. For a definite diagnosis of sarcopenic dysphagia, it is recommended to use explicit diagnostic criteria that includes the presence of dysphagia, low skeletal muscle strength and muscle mass, no obvious causative diseases of dysphagia, and low strength and mass of the swallowing muscles [8,11].

In this study, the degree of comorbidity burden according to the age adjusted CCI was relatively high and was not related to signs of dysphagia. Melgaard et al. [7] and Mañas-Martínez et al. [45] reached similar results, whereas Olesen et al. [6] and da Silva et al. [46] found that the presence of dysphagia was significantly related to higher scores of CCI. However, comorbidity is a complex concept, and it has been found that the CCI is only modestly associated with physical function [47]. Very few studies have considered the relationship between dysphagia and the state of chronic inflammation. Homem et al. [48] found a significant difference between patients with and without dysphagia regarding CRP, but not IL-6 and TNF-α levels. In the present study, elevated levels of four inflammatory biomarkers were present in the whole sample. However, at 56-week follow-up, the level of TNF-α was significantly higher in the group of patients with signs of dysphagia. In addition, signs of dysphagia at 56-week were associated with increased baseline levels of suPAR and IL-6. Since the level of comorbidity and inflammatory biomarkers was relatively high in the FAM-CPH cohort and there was a strong and significant association of signs of dysphagia and the three sarcopenia-related parameters at 56-week follow-up, it cannot be excluded that the presence of signs of dysphagia at 56-week follow-up was influenced by sarcopenia. Chronic inflammation affects the aging body with multiple impairments, for example hormonal and/or epigenetic alterations, microvascular changes, or insulin dysregulation, which may coalesce promoting sarcopenia [49]. In a cohort of patients with type 2 diabetes, Kaji et al. [50] observed that whole body sarcopenia was present in about 12% and was associated with decreased tongue strength, which might indicate that the condition of decreased insulin sensitivity led to low tongue strength. However, the biology of sarcopenia is complex and involves an intricate relationship between several pro- and anti-inflammatory proteins [49]. Moreover, the complex relationship between dysphagia and sarcopenia, both primary and secondary, and how sarcopenic dysphagia should be diagnosed remains unsolved, and further research is needed [8,11].

The present study found no association between signs of dysphagia and age or cognitive function, in contrasts to what has been demonstrated in previous research [7,15]. Since the FAM-CPH study excluded patients with severe cognitive impairments and patients who were loss to follow-up were significantly older, it cannot be excluded that cognitive function and age would have influenced the dysphagia status. Consistent with a growing body of evidence from various settings [1,5,6,7,12,13,18,43,51], signs of dysphagia in older patients were related to reduced swallowing activity, functional performance, and nutritional status. This persisted over time after discharge from the ED department. Unfortunately, the FAM-CPH data did not include information on patients’ rehabilitation plan. It has been found that in older patients with whole body sarcopenia undergoing rehabilitation for orthopedic diseases/conditions, adequate energy, and protein intake were associated with increased swallowing function and tongue strength [52]. Dysphagia and sarcopenia are both geriatric syndromes that isolated and in cooccurrence have a significant impact on older adults, resulting in poor outcomes [1,8,11,17,53]. The results from the FAM-CPH cohort provides support for the importance of systematic identification of sarcopenic dysphagia in at risk populations [1,8,11,17] such as older patients acutely admitted to an ED department. Timely identification of dysphagia would enable initiation of early and appropriate multidisciplinary rehabilitation strategies with nutritional care, strength training of the swallowing musculature, early mobilization, and physical exercise to improve or prevent the development of sarcopenic dysphagia in this patient cohort [8,11]. In addition, such an approach will provide detailed information for the rehabilitation plans across health care sectors ensuring quality and efficiency in the process for the acute patient.

This study is not without limitations. Application of secondary analysis on an existing dataset had some disadvantages as the data were not collected for addressing the aim of this study per se, and some study specific variables may have been left out [54]. Studies report that hospitalization may accelerate the development of sarcopenic dysphagia [11,13]. In present secondary analysis, patients who completed at 56-week follow-up had significantly lower LOS than patients lost to follow-up; and a post hoc analysis revealed that there were no significant differences in LOS between patients with or without signs of dysphagia at all three time-points. However, to determine the development of sarcopenic dysphagia after an ED-admission, a larger cohort without dysphagia at baseline would be needed. Besides, more comprehensive, and objective information on swallowing function, sarcopenia, as well as number and LOS of readmissions are required. The dataset used in present secondary analysis contained a relatively high magnitude of missing values. At 56-week follow-up, 48% (*n* = 60) were loss and there was > 20% missing values for the 30-CST and 4MGS at baseline. Accordingly, the power of the statistical analyses and the validity of the results may be limited. Additionally, it was not possible to investigate whether the course of signs of dysphagia were influenced by rehabilitation strategies since the available dataset did not provide information on the patients’ rehabilitation plans after the ED-visit. A significant limitation of present secondary analysis was that information on patients’ muscle mass and quality was not collected as part of the FAM-CPH study. Therefore, it was only possible to confirm the presence of probable sarcopenia and not a definite diagnosis of sarcopenia, according to the EWGSOP2 [9]. This also limited a diagnosis of sarcopenic dysphagia according to suggested diagnostic criteria [8,11]. Thus, further investigation in a larger sample with measures of all sarcopenia-related parameters (i.e., muscle strength, muscle quantity or quality and physical performance), comprehensive clinical examination of dysphagia (i.e., assessment of oral and motor function and functional oral intake) and identification of possible activity-, nutritional-, and disease-related risk factors for concurrent sarcopenia and dysphagia is required to fully understand the effects of sarcopenia on the development and/or progression of dysphagia in older patients acutely admitted to an ED.

## 5. Conclusions

Signs of dysphagia are prevalent in older patients acutely admitted to the ED, persists after discharge, and are associated with probable sarcopenia, inactivity, malnutrition, and chronic inflammation, both temporary and persistently over time. Sarcopenia-related parameters such as low muscle strength and physical performance as well as inactivity of swallowing, low functional performance status and malnutrition may be potentially modifiable targets for the improvement or prevention of dysphagia in older patients upon admission and after discharge from the ED. However, additional studies on sarcopenic dysphagia in older ED patients are warranted to establish definitive prevalence estimates, time course, and predictors.

## Figures and Tables

**Figure 1 geriatrics-06-00046-f001:**
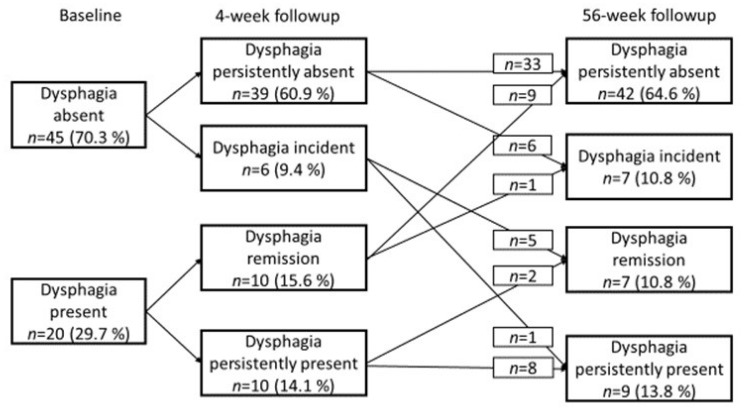
Course of dysphagia during a 56-weeks follow-up period.

**Table 1 geriatrics-06-00046-t001:** Characteristics of the FAM-CPH cohort

Variables	Total Baseline *n* = 125	Completed 56-Week Follow-Up *n* = 65	Loss 56-Week Follow-Up *n* = 60	*p*
Demographics				
Age in years ^a^	78.6 (8.3)	76.7 (7.8)	80.7 (8.4)	0.007
Female ^b^	70 (56.0%)	35 (53.8%)	35 (58.3%)	0.745
Male ^b^	55 (44.0%)	30 (46.2%)	25 (41.7%)	
Living situation *				
Together ^b^	43 (34.4%)	24 (36.9%)	19 (31.7%)	0.667
Alone ^b^	82 (65.6%)	41 (63.1%)	41 (68.3%)	
Housing				
Private residence ^b^	110 (88.0%)	59 (93.7%)	51 (85.0%)	0.205
Institution ^b^	13 (10.4%)	4 (6.3%)	9 (15.0%)	
Cognition				
OMC total score ^c^	24 (20; 26)	24 (22; 26)	18 (22; 26)	0.064
Swallowing function				
EAT-10 total score ^c^	0 (0; 3)	0 (0; 3)	0 (0; 4)	0.231
Parameters for probable sarcopenia				
HGS male (kg) ^a^	30.9 (10.4)	33.5 (10.1)	27.9 (10.0)	0.010
HGS female (kg) ^a^	17.7 (5.9)	18.9 (5.7)	16.5 (6.0)	0.098
30-CST^c^	0 (0; 10)	8 (0; 12)	0 (0; 5)	<0.001
4MGS (m/s) ^a^	0.70 (0.3)	0.8 (0.3)	0.6 (0.2)	0.001
Activity status				
SNAQ total score ^c^	14 (12; 16)	15 (13; 16)	14 (12; 16)	0.164
FRS total score ^c^	88 (77; 99)	98 (84; 100)	84 (69; 90)	<0.001
Nutritional status				
BMI (kg/m^2^) ^a^	26.2 (5.6)	26.2 (4.9)	26.3 (6.2)	0.894
MNA-SF total score ^c^	11 (8; 14)	12 (9; 14)	10 (7; 12)	0.019
Reasons for admission				
COPD exacerbation	13 (10.7%)	N/A	N/A	N/A
Dyspnea	26 (21.3%)	N/A	N/A	N/A
Pneumonia	7 (5.7%)	N/A	N/A	N/A
General symptoms & signs	28 (23.0%)	N/A	N/A	N/A
Chest pain, unspecified	23 (18.9%)	N/A	N/A	N/A
Other causes	25 (20.5%)	N/A	N/A	N/A
Disease status				
CCI ^c^	5.0 (3.0; 7.0)	5.0 (2.3; 6.0)	4.0 (6.0; 7.0)	0.019
CRP (mg/L) ^c^	22.0 (5.1; 75.0)	8.7 (2.5; 45.0)	48.5 (3.3; 177.5)	<0.001
suPAR (ng/mL) ^c^	4.4 (3.1; 6.2)	3.3 (2.8; 5.3)	4.9 (3.9; 6.5)	0.001
TNF-α (pg/mL) ^c^	12.5 (8.4; 17.7)	10.1 (7.2; 13.8)	15.0 (10.4; 19.8)	<0.001
IL-6 (pg/mL) ^c^	4.3 (1.6; 14.0)	2.1 (0.9; 6.3)	7.8 (3.2; 20.8)	<0.001
LOS (days) ^c^	3.1 (1.0; 7.2)	1.8 (0.8; 4.7)	5.7 (2.0; 9.7)	<0.001

Notes: * Institution includes nursing home, community-based rehabilitation center and senior residence. ^a^ Mean (SD); independent *t*-test; ^b^
*n* (%), Pearson **χ^2^** test with continuity correction or Fischer exact test; ^c^ Median (Q1; Q3), Mann–Whitney U test. N/A, not applicable. Significant level is set at 5%.

**Table 2 geriatrics-06-00046-t002:** Demographics, parameters for sarcopenia, activity, nutritional, and disease status according to signs of dysphagia by an EAT-score ≥ 2 points at three timepoints.

Variables	Dysphagia at Baseline			Dysphagia at 4-Week Follow-Up			Dysphagia at 56-Week Follow-Up		
	Absent *n*= 83 (66%)	Present *n*= 42 (34%)	*p*	Absent *n* = 70 (76%)	Present *N* = 22 (24%)	*p*	Absent *n* = 49 (75%)	Present *n* = 16 (25%)	*p*
Age ^a^	78.7 (8.7)	78.5 (7.3)	0.979	77.9 (7.9)	80.1 (10.1)	0.399	76.1 (7.9)	78.6 (7.2)	0.156
Female ^b^	43 (51.8%)	27 (64.3%)	0.184	39 (55.7%)	12 (54.5%)	0.923	26 (53.1%)	9 (56.3%)	0.824
Male ^b^	40 (48.2%)	15 (35.7%)		31 (44.3%)	10 (45.5%)		23 (46.9 %)	7 (43.8%)	
OMC-score ^c^	24 (20; 26)	22 (19; 25)	0.368	26 (22; 28)	24 (20; 28)	0.624	26 (22; 28)	26 (20; 28)	0.941
Parameters for probable sarcopenia									
HGS ^b^ ↔	60 (74.1%)	21 (51.2%)	0.020	48 (71.6%)	11 (55.0%)	0.260	39 (83.0%)	9 (56.3%)	0.043
HGS ^b^ ↓	21 (25.9%)	20 (48.8%)		19 (28.4%)	9 (45.0%)		8 (17.0%)	7 (43.8%)	
30-CST ^b^ ↔	27 (42.2%)	7 (20.0%)	0.029	46 (68.7%)	13 (38.1%)	0.024	37 (78.7%)	5 (31.2%)	0.002
30-CST ^b^ ↓	37 (57.8%)	28 (80.0%)		21 (31.3%)	8 (61.9%)		10 (21.3%)	11 (68.8%)	
4MGS ^b^ ↔	16 (26.7%)	5 (20.0%)	0.591	28 (58.2%)	6 (33.3%)	0.704	27 (58.7%)	3 (20.0%)	0.021
4MGS ^b^ ↓	44 (73.3%)	20 (80.0%)		39 (41.8%)	12 (66.7%)		19 (41.3%)	12 (80.0%)	
Activity status									
SNAQ score ^c^	15 (3; 16)	13 (11; 16)	0.005	15 (14; 16)	14 (13; 15)	0.022	16 (14; 16)	14 (12; 15)	<0.001
Swallow activity ^b^ ↔	56 (69.1%)	19 (45.2%)	0.012	53 (75.7%)	11 (50.0%)	0.043	42 (89.4%)	10 (62.5%)	0.024
Swallow activity ^b^ ↓	25 (30.9%)	23 (54.8%)		17 (24.3%)	11 (50.0%)		5 (10.6%)	6 (37.5%)	
FRS score^c^	92 (80; 99)	86 (58; 92)	0.005	94 (78; 100)	77 (56; 99)	0.035	98 (88; 100)	75 (36; 90)	0.001
Nutritional status									
BMI(kg/m^2^) ^a^	27.1 (5.5)	24.6 (5.3)	0.016	26.6 (5.4)	22.9 (6.0)	0.007	27.6 (5.0)	23.1 (4.4)	0.002
MNA-SF score^c^	11 (9; 14)	9 (7; 12)	0.004	11 (9; 12)	9 (6; 12)	0.117	14 (12; 14)	11 (8; 13)	0.002
Normal ^b^	30 (37.0%)	9 (21.4%)	0.009	30 (43.5%)	7 (31.8%)	0.185	38 (79.2%)	7 (43.8%)	0.012
At risk ^b^	34 (42.0%)	13 (31.0%)		29 (42.0%)	8 (36.4%)		10 (20.8%)	7 (43.8%)	
Malnutrition ^b^	17 (21.0%)	20 (47.6%)		10 (14.5%)	7 (31.8%)		0 (0.0%)	2 (12.4%)	
Disease status									
CCI ^c^	5.0 (3.0; 7.0)	5.0 (3.0; 7.0)	0.947	NC	NC		NC	NC	
CRP (mg/L) ^c^	23.0 (3.8; 81.0)	20.5 (4.7; 54.0)	0.967	3.3 (1.3; 8.8)	14.0 (5.2; 18.5)	0.547	1.9 (1.0; 4.7)	5.2 (1.6; 11.3)	0.128
suPAR (ng/mL) ^c^	4.6 (3.1; 6.3)	4.0 (3.0; 5.8)	0.337	3.6 (2.7; 5.3)	4.7 (3.1; 5.9)	0.928	3.1 (2.5; 3.8)	3.2 (2.3; 5.9)	0.790
TNF-α (pg/mL) ^c^	12.0 (8.2; 17.5)	12.6 (8.7; 17.9)	0.739	9.8 (7.7; 14.4)	13.9 (9.1; 17.8)	0.260	9.1 (6.9; 11.9)	11.4 (9.6; 14.4)	0.039
IL-6 (pg/mL) ^c^	4.4 (1.4; 14.3)	3.7 (2.1; 12.7)	0.780	1.3 (0.7; 2.3)	4.1 (1.7; 15.9)	0.745	1.0 (0.6; 1.5)	1.4 (0.8; 2.7)	0.081

Notes: ^a^ Mean (SD), independent *t*-test; ^b^
*n* (%), Pearson **χ^2^** test with continuity correction or Fischer exact test; ^c^ Median (Q1; Q3), Mann–Whitney U test; ↔ ( = intact according to sarcopenia cut-off point (9,29)); ↓ ( = low according to sarcopenia cut-off point (9,29)); NC, not computed since CCI is only calculated for baseline data; significant level is set at 5%.

**Table 3 geriatrics-06-00046-t003:** Association of signs of dysphagia with parameters for probable sarcopenia and activity, nutritional, and disease status at three time points

	Baseline		4-Week Follow-Up		56 Week Follow-Up	
Parameters for probable sarcopenia	Crude OR (95%CI)	*p*	Crude OR (95%CI)	*p*	Crude OR (95%CI)	*p*
Low handgrip strength	2.72 (1.23; 5.99)	0.013	2.07 (0.74; 5.78)	0.167	3.79 (1.09; 13.19)	0.036
(HGS < 27 kg (men)/<16 kg (women))						
Low leg strength and endurance (30-CST < 9 rises)	2.92 (1.11; 7.67)	0.030	3.56 (1.28; 9.88)	0.015	8.14 (2.29; 28.90)	0.001
Low physical performance (4MGS ≤ 0.8 m/s)	1.38 (0.44; 4.32)	0.578	1.44 (0.48; 4.29)	0.517	5.68 (1.41; 22.93)	0.015
Activity status						
SNAQ score (swallow activity)	0.81 (0.70; 0.96)	0.004	0.82 (0.66; 1.02)	0.075	0.56 (0.39; 0.80)	0.002
FRS score (Functional performance status)	0.98 (0.96; 0.99)	0.019	0.98 (0.96; 1.00)	0.048	0.96 (0.93; 0.99)	0.002
Nutritional status						
MNA-SF score	0.82 (0.71; 0.94)	0.005	0.87 (0.74; 1.01)	0.072	0.64 (0.47; 0.86)	0.003
BMI (kg/m^2^)	0.92 (0.85; 0.99)	0.019	0.87 (0.79; 0.97)	0.010	0.79 (0.67; 0.93)	0.005
Disease status						
CCI	1.00 (0.87; 1.14)	0.985	NC		NC	
CRP (mg/L)	1.00 (0.99; 1.00)	0.594	1.00 (0.98; 1.03)	0.792	1.11 (0.98; 1.3)	0.089
suPAR (ng/mL)	0.90 (0.76; 1.07)	0.239	0.97 (0.76; 1.24)	0.796	1.15 (0.87; 1.52)	0.323
TNF-α (pg/mL)	0.98 (0.94; 1.02)	0.421	1.01 (0.95; 1.06)	0.822	1.13 (1.00; 1.28)	0.057
IL-6 (pg/mL)	1.00 (0.99; 1.00)	0.546	1.07 (0.95; 1.19)	0.274	1.16 (0.94; 1.44)	0.163

Note: NC, not computed since CCI is only calculated for baseline data; significant level is set at 5%.

**Table 4 geriatrics-06-00046-t004:** Associations of signs of dysphagia at 56-week follow-up and baseline status.

Univariate Logistic Regression		
	Crude OR (95% CI)	*p*
Parameters for probable sarcopenia at baseline		
Low handgrip strength (HGS < 27 kg (men)/<16 kg (women))	2.96 (0.88; 9.90)	0.079
Low leg strength and endurance (30-CST < 9 rises)	4.82 (1.16; 19.99)	0.030
Low physical performance (4MGS ≤ 0.8 m/s))	6.77 (0.80; 57.5)	0.080
Activity status at baseline		
Swallowing activity (SNAQ total score)	0.71 (0.55; 0.93)	0.011
Low swallowing activity (SNAQ < 14 point)	5.61 (1.66; 19.90)	0.005
Functional performance status (FRS total score)	0.95 (0.92; 0.98)	0.001
Nutritional status at baseline		
Nutritional status (MNA-SF total score)	0.68 (0.54; 0.86)	0.001
Low Nutritional status (MNA-SF, at risk or malnourished)	4.35 (1.23; 15.44)	0.023
Nutritional status (BMI (kg/m^2^)	0.87 (0.76; 0.99)	0.038
Disease status at baseline		
CCI	1.02 (0.84; 1.25)	0.828
CRP (mg/L)	1.01 (1.00; 1.02)	0.204
SuPAR (ng/mL)	1.33 (1.02; 1.73)	0.035
TNF-α (pg/mL)	1.12 (1.00; 1.26)	0.060
IL-6 (pg/mL)	1.08 (1.01; 1.15)	0.035

Note: Significant level is set at 5%.

## Data Availability

Data available on request due to restrictions. The data presented in this study are not publicly available due to Danish legislation. Request to access the dataset will require an individual inquiry to the Danish Data Protection agency for approval.

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
