# Peer review of "Dysphagia Prevalence, Time Course, and Association with Probable Sarcopenia, Inactivity, Malnutrition, and Disease Status in Older Patients Admitted to an Emergency Department: A Secondary Analysis of Cohort Study Data"

_geriatrics, 2021, doi:10.3390/geriatrics6020046_

Round 1

Reviewer 1 Report

This is a prospective, longitudinal observational cohort study aimed to investigate prevalence and time-course of dysphagia in older patients admitted to an emergency department as well as its association with sarcopenia and sarcopenia-related conditions. This topic is interesting for research and the methods are well design and implemented but it presents an important limitation related to sarcopenia diagnosis because the EWGSOP2 algorithm for detecting and diagnosing cases of sarcopenia has not been applied in its entirety. The lack of a quality or quantity of muscle mass variable leads to talk of probable sarcopenia but does not allow the diagnosis of sarcopenia to be established.  This conditions to speak throughout the text of sarcopenia, so it would be advisable to speak of probable sarcopenia. I would like to mention the following changes/suggestions for being considered. Title- I think that the title could improve in this way: “Dysphagia prevalence, time course and association with probable sarcopenia and related conditions in older patients admitted…”.

Abstract-      Conclusion has been rewritten according to the title “probable sarcopenia”. Introduction-      Please, rewrite the aim according title. Methods- The EWGSOP2 consensus says “…detection of low muscle quantity and quality to confirm the sarcopenia diagnosis…” (Cruz-Jentoft et al 2019). This information is not included in section 2.2.3-      I don’t understand as sarcopenia-related condition the swallowing activity and functional status.  

Results

- In tables 2-3 there are errors and missing information. For example, in table 2 no correct p-values in 4MGS (059) swallow activity no visible number, incomplete sentence in notes section…

Discussion and conclusion

- I find your study and the obtained results very interesting, but I consider a great weakness that you have not collected muscle mass when the main outcome is sarcopenia.

Author Response

Response to Reviewer 1 Comments

Point 1: This topic is interesting for research and the methods are well design and implemented but it presents an important limitation related to sarcopenia diagnosis because the EWGSOP2 algorithm for detecting and diagnosing cases of sarcopenia has not been applied in its entirety. The lack of a quality or quantity of muscle mass variable leads to talk of probable sarcopenia but does not allow the diagnosis of sarcopenia to be established. This conditions to speak throughout the text of sarcopenia, so it would be advisable to speak of probable sarcopenia. I would like to mention the following changes/suggestions for being considered. Title- I think that the title could improve in this way: “Dysphagia prevalence, time course and association with probable sarcopenia and related conditions in older patients admitted…”.

Response 1: We do agree that a major limitation of this secondary analysis is that we could not confirm the diagnosis of sarcopenia since the information on muscle quantity and quality was not provided in the data set from the FAM-CPH study. This is discussed in the original manuscript. Nevertheless, we agree that it is appropriate to be more explicit on this issue. In the original manuscript (line 159-161), we formulated that a definite diagnosis of sarcopenia as defined by EWGSOP2 [9] was not possible, and the three included sarcopenia parameters were considered individually as signs of probable sarcopenia. As suggested, we have revised the term sarcopenia into ‘probable sarcopenia’ throughout the manuscript text and tables (highlighted with yellow in the revised manuscript) and we have changed the tittle according to the suggestion made by reviewer 1. In fact, we have also considered the comment made by Reviewer 1 ‘ I don’t understand as sarcopenia-related condition the swallowing activity and functional status’.  (see response to point 5)

Therefore, the title is edited into ‘Dysphagia prevalence, time course and association with probable sarcopenia, inactivity, malnutrition and disease status in older patients admitted to an emergency department: a secondary analysis of cohort study data’

Point 2: Abstract- Conclusion has been rewritten according to the title “probable sarcopenia”.

Response: We have re-written as suggested ‘Signs of dysphagia in older patients admitted to an ED was prevalent, persisted 56 weeks after discharge, and was associated with probable sarcopenia and related conditions; all modifiable targets for management of dysphagia in older patients’.

Point 3: Introduction- Please, rewrite the aim according title.

Response: We have rewritten the aim according to the revised title to ‘the aim of the present study was to investigate the prevalence and time course of dysphagia and its association with parameters of sarcopenia, inactivity, malnutrition, disease status and systemic inflammation in older patients acutely admitted to an ED within the Danish health care system’ (revised manuscript line 94-97).

Point 4: Methods- The EWGSOP2 consensus says “… detection of low muscle quantity and quality to confirm the sarcopenia diagnosis…” (Cruz-Jentoft et al 2019). This information is not included in section 2.2.3-

Response: We have added the following in the method section ‘According to the EWGSOP2 [9], a confirmed diagnosis of sarcopenia includes evidence of low muscle quality and quantity (revised manuscript line 162-163).

Point 5: I don’t understand as sarcopenia-related condition the swallowing activity and functional status.

Response: We have decided not to use the phrase ‘sarcopenia-related conditions. Instead, we simply, uses the terms activity-, nutrition- and disease status. These variables are known as consequences of dysphagia and as contributors for secondary (or iatrogenic) sarcopenia and is categorized into activity-related, nutrition-related, and disease-related-iatrogenic sarcopenia. However, since we were not able to confirm the diagnosis of sarcopenia and it appears that the term night led to confusion, we believe that is it more appropriate not to use the term sarcopenia-related conditions, but the conditions them self.

Point 6: Results - In tables 2-3 there are errors and missing information. For example, in table 2 no correct p-values in 4MGS (059) swallow activity no visible number, incomplete sentence in notes section…

Response: We really do regret all the errors in table 2 and 3, which might have been a result of the submission process. It seems that the layout in our original manuscript have been changed resulting in a different format of the tables. Apparently, the consequences were that a vast majority of the tabulated values were hidden. We believe that we have solved all the problems for table 2 (page 8) and table 3 (page 9).

Point 7: Discussion and conclusion - I find your study and the obtained results very interesting, but I consider a great weakness that you have not collected muscle mass when the main outcome is sarcopenia.

Response: We utterly agree that it is a significant limitation that the available dataset used in our secondary analysis did not include information on muscle mass. And we are aware that application of secondary analysis on an existing dataset had some disadvantages as the data were not collected for addressing the aim of this study per se. Therefore, as stated on line 115-117, the primary outcome was dysphagia, and not sarcopenic dysphagia; and the parameters for probable sarcopenia, activity-, nutrition- and disease- status were kept as secondary outcomes. However, to emphasize the named limitation, we have elaborated our discussion of the limitation of this study (line 393-404) as follows:

‘A significant limitation of present secondary analysis was that information on patients’ muscle mass and quality was not collected as part of the FAM-CPH study. Therefore, it was only possible to confirm the presence of probable sarcopenia and not a definite diagnosis of sarcopenia, according to the EWGSOP2 [9]. This also limited a diagnosis of sarcopenic dysphagia according to suggested diagnostic criteria [8,11]. Thus, further investigation in a larger sample with measures of all sarcopenia-related parameters (i.e., muscle strength, muscle quantity or quality and physical performance), comprehensive clinical examination of dysphagia (i.e. assessment of oral and motor function and functional oral intake) and identification of possible activity- nutritional- and disease-related risk factors for concurrent sarcopenia and dysphagia is required to fully understand the effects of sarcopenia on the development and/or progression of dysphagia in older patients acutely admitted to an ED’.

Reviewer 2 Report

In this study, the authors have demonstrated that signs of dysphagia in older patients admitted to an emergency department was prevalent, persisted 56 weeks after discharge, and was associated with sarcopenia parameters and sarcopenia-related conditions. This reviewer raises few points that have to be addressed by authors.

  1. Sarcopenia parameters and state of chronic inflammmation may also be dependent on another underlying factors such as hypertension, diabetes or liver function (with or without NAFLD). Please clarify a little more about them.
  2. Following article will be helpful.

Kaji A, et al. Sarcopenia is associated with tongue pressure in older patients with type 2 diabetes: A cross-sectional study of the KAMOGAWA-DM cohort study. Geriatr Gerontol Int. 2019 Feb;19(2):153-158. doi: 10.1111/ggi.13577.

Author Response

Response to Reviewer 2 Comments

Point 1. Sarcopenia parameters and state of chronic inflammation may also be dependent on another underlying factors such as hypertension, diabetes or liver function (with or without NAFLD). Please clarify a little more about them.

Point 2: Following article will be helpful. Kaji A, et al. Sarcopenia is associated with tongue pressure in older patients with type 2 diabetes: A cross-sectional study of the KAMOGAWA-DM cohort study. Geriatr Gerontol Int. 2019 Feb;19(2):153-158. doi: 10.1111/ggi.13577.

Response to point 1 & 2 as they are related: Thank you for attending us to the work by Kaji and colleagues. We do agree that sarcopenia and chronic inflammation may be dependent on several impairments and diseases. And that there might be e reverse relationship as well.  We have addressed this issue more explicit in the revised manuscript (line 342-351) as ‘Chronic inflammation affects the aging body with multiple impairments, for example hormonal and/or epigenetic alterations, microvascular changes, or insulin dysregulation, which may coalesce promoting sarcopenia [49]. In a cohort of patients with type 2 diabetes, Kaji et al [50] observed that whole body sarcopenia was present in about 12% and was associated with decreased tongue strength, which might indicate that the condition of decreased insulin sensitivity led to low tongue strength. However, the biology of sarcopenia is complex and involves an intricate relationship between several pro- and anti-inflammatory proteins [49]. Moreover, the complex relationship between dysphagia and sarcopenia, both primary and secondary, and how sarcopenic dysphagia should be diagnosed remains unsolved, and further research is needed [8,11].

Round 2

Reviewer 1 Report

I thank the authors for responding in detail to the questions raised, and for improving the writing of the manuscript with the changes made.

Please, review table 2 because there are some errors in numbers (p.e: 4MGS; p=0591). Tables 2, 3 and 4 include some abbreviations that are not detailed.

Author Response

Thank you for a very constructive review. We are pleased that our revisions were satisfactory. Below you find our responses to the raised minor revisions. The revisions from this second round are highlighted in green.

Point 1: Please, review table 2 because there are some errors in numbers (p.e: 4MGS; p=0591).

Response 1: We have reviewed all tabulated values in table 2 and have corrected one identified error which is identical with the one pointed out (highlighted in green in table 2). In addition, we have also reviewed all the other tables, and have not identified more errors.

Point 2: Tables 2, 3 and 4 include some abbreviations that are not detailed.

Response 2: We have reviewed all abbreviations in table 2,3 and 4, and all are defined in the text of the method sections. In agreement with Assistant Editor Antonela Cazacu, definitions of these abbreviations are therefore omitted in notes below the tables. However, the units of measurement in terms of m/s for 4MGS and the standard units of measurements for the biomarkers were not defined. Although, these are recognized standard units, we have added their definitions in the method section line 160 and line 195-197 (highlighted in green).

Be aware that the Track Changes function in words results in change of the format, especially for the tables, where some values become hidden. Therefore, we have also uploaded a pdf version of the revised manuscript, in which only highlights with colors appear (1 round review: yellow and 2 round review: green). Alternatively, it helps highlighting the table and accept the changes. Then all values become visible.
